# Welcome to Canada: Why Are Family Emergency Shelters 'Home' for Recent Newcomers?

**Katrina Milaney** [1,*] , **Rosaele Tremblay** [1] , **Sean Bristowe** [1] **and Kaylee Ramage** [2]

1   Community Health Sciences, Cumming School of Medicine, Calgary, AB T2N 4Z6, Canada;
    tremblar@ucalgary.ca (R.T.); sean.bristowe@ucalgary.ca (S.B.)
2   Community Health Sciences, Obstetrics and Gynecology, Cumming School of Medicine,
    Calgary, AB T2N 4Z6, Canada; kaylee.ramage@ucalgary.ca
*   Correspondence: katrina.milaney@ucalgary.ca

**Abstract:** Although Canada is recognized internationally as a leader in immigration policy, supports are not responsive to the traumatic experiences of many newcomers. Many mothers and children arriving in Canada are at elevated risk of homelessness. **Methods:** This study utilized a community-engaged design, grounded in a critical analysis of gender and immigration status. We conducted individual and group interviews with a purposive sample of 18 newcomer mothers with current or recent experiences with homelessness and with 16 service providers working in multiple sectors. **Results:** Three main themes emerged: gendered and racialized pathways into homelessness; system failures, and pre- and post-migration trauma. This study revealed structural barriers rooted in preoccupation with economic success that negate and exacerbate the effects of violence and homelessness. **Conclusion:** The impacts of structural discrimination and violence are embedded in federal policy. It is critical to posit gender and culturally appropriate alternatives that focus on system issues.

**Keywords:** immigration; mothers; homelessness; structural violence

## 1. Introduction

Globally, Canada is recognized as a welcoming country for newcomers and has a strong reputation as an international leader in inclusive immigration practices [1,2]. Canada's federal immigration policy, the *Immigration and Refugee Protection Act*, SC 2001, c 27 (The Act) aims to encourage economic growth, "support the development of a strong and prosperous Canadian economy", "permit Canada to pursue the maximum social, cultural and economic benefits of immigration", "to offer safe haven to persons with a well-founded fear of persecution based on race, religion, nationality, political opinion or membership in a particular social group", to reunite families, and offer a safe place for those fleeing conflict and violence [3,4].

In 2015, over 270,000 [3] people came to Canada and the numbers are forecasted to rise by 40% in 2020 [1]. Canada's economy depends on the continuous flow of newcomers for economic growth and allows newcomers to come under many classes, including economic, sponsorship, and protected persons. *Economic* class allows individuals to arrive in Canada based on specific skills that are beneficial and valuable for economic growth [4]. The main applicant can apply under this category, while their spouse or partner applies as a dependent [3]. More men (60.6%) than women enter Canada in this category. *Sponsorship* allows for a Canadian citizen or permanent resident to take on the legal and financial responsibility of supporting a newcomer. Sponsorship in Canada can occur for three groups: dependent children, spouses or common-law partners; or families [3,4]. In 2015, 24.1% arrived within these categories [5]. *Protected Persons or refugee* class accepts individuals who require protection because

their life is in danger and/or they are fleeing torture or punishment [4,6]. A total of 32,115 refugees and protected persons arrived in Canada in 2015 [5].

Priority is given to those who enter in the 'economic class' "*in 2017, the Government of Canada adopted a historic multi-year levels plan to responsibly grow our annual immigration levels … by 2020, with 60 percent of the growth in the Economic Class. Growing immigration levels, particularly in the Economic Class, will help us sustain our labour force, support economic growth and spur innovation.*" [5] (p. 2).

Federal policy mandates that newcomers receive supports for housing and other basic needs for up to one year; however, many immigrant and refugee families are dependent upon, yet unable to access, these supports [7] Data from two family emergency shelters in a large Canadian urban center indicate that almost 30% of families on any given day are newcomers [8]. Lone-parent women and children make up the majority of families experiencing homelessness and experience high rates of family violence. This creates complexities to providing safe and holistic responses meant to support the whole family [9,10].

Understanding the complex experience of homelessness for newcomer mothers requires an examination of social context, including the ways in which newcomers are classified and the subsequent supports available to them, as well as the confluence of pre- and post-migration traumas they face. Critically examining this phenomenon requires an understanding that neoliberalism, or the hegemonic political ideology in Canada, has influenced federal immigration policy [11]. An intersectional analysis rooted in gender and culture reveals that the impact of federal policy on many newcomer women and children is structural violence and discrimination.

Structural violence is defined as the violence occurring when social structures or institutions cause harm by preventing individuals from meeting their basic needs [12,13]. Structural violence and interpersonal violence are interdependent and interwoven. Racial violence and gendered violence, for instance, do not exist in isolation of social structures that create inequity. Structural violence manifests as unequal access to social systems of support like housing, health care, education and employment, primarily due to a long history of cuts to social welfare funding and programs that disproportionately affect women. This in turn increases the risk for interpersonal violence, as women are 'forced' into vulnerable social positions and dependency [14]. Racism and discrimination are terms that are often used interchangeably—in this study, we choose to use the term discrimination as it has been defined as unequal treatment based on group membership. Structural discrimination refers to the "ideologies, practices, processes, and institutions that operate at the macro level to produce and reproduce differential access to power and to life opportunities along racial and ethnic lines" [15] (p. 2102). Like structural violence, structural discrimination is often embedded in policies, institutions and subsequent practices. Although not every newcomer to Canada is a member of a racial or ethnic minority, they may still experience the impacts of discrimination, which has been defined as "any distinction, exclusion, restriction or preference based on race, colour, descent, or national or ethnic origin" [16] (Article 1). The Canadian Council for Refugees (2000) notes that "in Canada, where open expression of racist ideas is generally not tolerated, hostility towards newcomers serves as an outlet for the expression of underlying racist sentiments" (para 2) [17]. Furthermore, limiting discussion of race and discrimination to situations where there are differences in genetically-determined, phenotypical characteristics between groups excludes situations where race is perceived to be an issue and where discrimination may still occur [18].

Limited research has focused on the gendered experience of newcomer homelessness in Canada through a structural lens. Although migrant homelessness has been studied in Canada [19], studies examining the specific factors affecting migrant women's homelessness [20,21] have focused on describing the experiences of women at the individual level. To address this gap in the literature and to examine the impact of Canadian immigration policy on newcomer women's homelessness, we conducted a qualitative research project with newcomers who experienced homelessness in a large Canadian city with a high rate of immigrant and refugee resettlement. The research questions

for this study were: What are the experiences of homelessness for newcomer women in Canada? How does Canadian immigration policy contribute to newcomer women's homelessness?

## 2. Background

### 2.1. Immigration in Canada and Calgary

Canada has a high population of immigrants, with almost 22% of the population reporting that they were or had been a landed immigrant or permanent resident in the most recent census [22]. In 2016, Canada had over 1.2 million new immigrants (i.e., those who had permanently settled in Canada in the last five years), representing 3.5% of the Canadian population. The majority of these new immigrants were admitted under the economic class (60.3%), 26.8% under the family class, and 11.6% as refugees. Although most new immigrants to Canada settle in the large metropolitan areas of Toronto (Ontario), Vancouver (British Columbia), or Montreal (Québec), Alberta is emerging as a desirable location for resettlement. From 2001 to 2016, the percentage of immigrants living in Alberta rose from 6.9% to 17.1% [22].

Although Canadian immigration policy is meant to promote resettlement, newcomers to Canada face many barriers to resettlement, including access to childcare [23], employment issues (e.g., devaluation of foreign credentials, lack of Canadian work experience) [24], access to health care [25], and access to stable and affordable housing [6]. Discrimination is also a common experience for newcomers, with discrimination based on ethnicity or culture, race, language, or physical appearance. A study of discrimination among immigrants in Canada found that 28% of immigrants had faced discrimination from a person in authority and 25% faced discrimination from a service provider [26].

### 2.2. Canadian Immigration Policy

The federal Ministry of Immigration, Refugees and Citizenship Canada (IRCC) is responsible for laws and polices pertaining to immigration.

"IRCC strengthens Canada's economic, social and cultural prosperity by helping ensure Canadian safety and security while managing one of the largest and most generous immigration programs in the world IRCC also conducts, in collaboration with its partners, the screening of potential permanent and temporary residents to protect the health, safety and security of Canadians. Fundamentally, the Department builds a stronger Canada by helping immigrants and refugees settle and fully integrate into Canadian society and the economy, and by encouraging and facilitating Canadian citizenship [5]."

Supports are available, depending on immigration class, and may include help accessing housing, finding employment and language programs [1]. However, a growing number of newcomers in Canada are considered to have precarious immigration status [27]. According to these researchers, precarious status "indicates the lack of any of the following: (1) work authorization, (2) the right to remain permanently in the country (residence permit), (3) social citizenship rights available to permanent residents (e.g., education and public health coverage), and (4) not depending on a third party for one's right to be in Canada (such as a sponsoring spouse or employee)" (p. 240–241). If any one member of the family has precarious status, other family members, including children, are at risk of "deep social exclusion that can contribute to negative social and health outcomes" [11] (p. 213). In 2009, nearly one million temporary residents or approximately one in thirty-four people in Canada have some form of precarious immigration status. Many women fall into this category, particularly if they entered Canada through marriage or spousal sponsorship because current federal legislation grants control over the immigration processes to the resident spouse [28]. Refugees often receive additional financial support during their first year in Canada [29]. However, the recent influx of refugees has overwhelmed programs, resulting in rushed and poor service delivery and long waitlists [30]. There is an inherent urgency in addressing these issues, not only in recognition of human dignity but also because of the children that are at increased risk of adverse intergenerational outcomes due to continuing instability, homelessness and subsequent trauma.

## 3. Trauma

In this study, trauma refers to the difficult events women may have experienced either fleeing their home country or when they arrived in Canada. Many refugees, in particular, have experienced trauma in varying forms of violence [5,31]. Many have also witnessed or been actively engaged in traumatic events while fleeing their home country.

Mental health issues like anxiety, depression and post-traumatic stress are common, and are directly linked to experiences such as losing a family member, witnessing or being subjected to violence or living in a dangerous war zone [31,32]. Women, in particular, may be victims of rape, harassment, and sexual slavery [33,34]. Men will often use sexual violence to control, humiliate and demoralize women in situations of war and strong military presence. These experiences affect individuals, families, and communities and their subsequent generations and depression, grief, and hopelessness are only some ways trauma affects victims [35–37].

Many mothers who immigrate with the support of a spouse encounter interpersonal violence (IPV) [38]. For some, IPV is cited as the reason for leaving their country (pre-migration trauma) [39–41]. The stressors of income insecurity, changing family dynamics and acculturation to Canadian customs can exacerbate an already stressful migratory experience [42]. Post-migration trauma is described as negative and distressing encounters with violence after migration or with Canada's institutions including immigration, health care and the social service sector [43].

The Canadian Council for Refugees states that newcomer women suffer unique forms of partner abuse because they may be unaware of their rights in Canada [44]. Abusive partners who are also sponsoring these women are gatekeepers to information on legal rights, available services, and the immigration process [44,45]. Fear of deportation or fear that the sponsor will withdraw their support or take their children also causes women to live in constant fear and stress [45].

### *Violence and Homelessness*

Immigrant and refugee mothers are at an elevated risk of homelessness and its adverse effects, with non-status refugee claimants spending the longest time in shelters [46]. Studies have shown that some women experience family violence for the first time once they arrive in Canada. This issue is called the "sponsorship effect", where the abusive partner is in control of the immigration process [37,47]. Newcomer women often apply for permanent residency as dependents, while their partner is the primary applicant [3,9]. This places women in a position of dependency on the applicant [48]. Immigrant and refugee men may struggle with resettlement and employment and may subsequently perceive a loss of dominant status in their family. Consequently, they may attempt to increase their power and control over their spouse and children [33]. Interpersonal violence is the most common reason for immigrant and refugee women to leave their partners [9,49]. Mothers may be forced into homelessness to escape an abusive relationship and to protect their children. However, living in a shelter or on the streets can lead to women losing custody of their children and fear of losing residency status in Canada [37]. Social exclusion manifests in some family members segregating or isolating those women who try to leave or speak out against family violence [9,37]. Furthermore, many services in domestic violence shelters are not available to newcomer women because of limitations to eligibility due to their immigration status and cuts to social services that have meant that violence shelters are not able to provide culturally and linguistically appropriate services to newcomer women [50].

For many newcomers, multiple structural factors affect their resettlement in Canada. Some of these issues, such as the affordable housing crisis in Canada, are not specific to newcomers, but have particular effects on newcomers often due to their vulnerability and dependency on public systems of support upon their arrival to Canada. Others are more unique to newcomers, such as policies dictating available supports based on their status in Canada, sponsorship rules, and wait times for court dates for obtaining permanent residency [51].

## 4. Neoliberalism

The usefulness of neoliberalism as a framework to study social inequities has been debated. Many scholars argue that the term is used vaguely and began as an economic approach that has 'morphed' into a catch-all phrase for a wide range of social problems, challenging its efficacy as a tool for analysis [52]. Others, particularly feminist scholars, debate the inextricable and dynamic relationship between feminism, neo-liberalism and capitalism as necessary to critically examine persistent inequities for women [53]. In this paper, we argue that examining current immigration practices and their subsequent impact on the daily lives of women should not be done in the absence of understanding dominant political and economic ideology—in particular, neoliberalist approaches preoccupied with economic independence and self-reliance that lead to cuts to social programs for newcomers and 'offloading' responsibility for 'successful' settlement to family members and/or non-governmental organizations (sponsors). Neoliberalist policies, focused on economic prosperity, have been argued to be at the root of Canada's building immigration 'crisis'. Federal plans to rapidly increase the number of newcomers to Canada to bolster economic prosperity exist within a shrinking social safety net [7].

Given the current global preoccupation with capitalist values—a resurgence of authoritarian capitalism, nationalism and right-wing ideology, led by the United States but influencing Westernized countries globally [54]—we argue an urgent need for a critical feminist, intersectional examination of the 'problem' of homelessness for newcomer families to help challenge this resurgence and create alternatives.

The objectives of this study were three-fold: firstly, to examine newcomer mothers' experiences of homelessness to identify structural imbalances and gaps; secondly, to understand how structural barriers influence violence and trauma; and finally, to recommend changes to both policy and service delivery to address these structural issues.

## 5. Methodology

We conducted a qualitative study in a large Canadian city (Calgary, Alberta) using Critical Social Theory (CST) as a methodological lens. CST is a particularly useful lens for the examination of complex social problems, as it attempts to connect individual issues to structural issues of power, exclusion, and ideology rather than treating problems as residing within the individual [55]. CST allows us to examine the multiple experiences of marginality and to challenge the structural barriers and power differentials that maintain inequities in service delivery and policy development [56]. Homelessness is one such issue, with complexities influenced by structural-, community-, and individual-level factors [9]. Applying a CST lens in our study required a consideration of the gendered experience of homelessness as different for women than it is for men [9,37], while simultaneously analyzing the nuanced experiences of newcomer women, because they are newcomers. Gender differences include women's pervasive experience of violence and having children in their care [47–49]. Gendered theory argues that motherhood is a socially determined role that can provide a deep sense of connection and meaning when society views motherhood as valuable [57]. However, mothering can also be experienced as an oppressive role for mothers in homelessness as the norms of a 'good mother' (i.e., having a stable marriage, education, income, housing and providing a safe environment for their children) are impossible to meet [58].

Understanding how cultural differences shape mothers' experiences helps illuminate the ways in which immigration policy and subsequent processes create structural barriers for mothers and their children [9,59]. Understanding that homelessness for newcomer mothers is rooted in gendered and cultural norms and expectations allows for an examination of responses that focus on challenging social systems and structures rather than individual-level issues that affect women who are often forced into dependency upon these systems. In this way, 'neutral' analyses, which ignore and negate differences and consequently create inequitable or ineffective responses, can be challenged. We argue that "a richer understanding of the structural determinants influencing women's trajectories into homelessness requires

an intersectional approach that considers the simultaneous and mutually constitutive effects of the multiple social categories of identity, difference, and disadvantage that individuals inhabit" [15] (p. 2009).

## 5.1. Critical Ethnography and Community-Engaged Research

Ethnographic research is a qualitative design that explores shared social experiences rather than testing a pre-determined hypothesis [60–62]. Wolcott [63] proposes ethnography as an appropriate methodology to use when addressing questions focusing on the cultural context of a specific group. Ethnographic studies tend to focus on small samples to allow for a more in-depth study of their experiences [61]. It should be noted, however, that throughout history, ethnographic methodology has been criticized for engaging in unethical and racist practices that further marginalize the groups being studied [64]. Consequently, it was imperative that this study approached the design and participants with gender and culturally safe methods. As such, this project applied a community-engaged research design (CER), guided by the collection methods of ethnography [65]. CER is a unique method for "scientific discovery bringing transdisciplinary teams together that study problems in real-world contexts" [66] (p. 93). Our approach included three representatives from a family emergency shelter as part of the research team. In addition, we engaged 24 representatives from 16 community-based organizations in the homeless, housing, and immigrant sectors as a Community Advisory Committee (CAC). These representatives were recruited on a volunteer basis from relevant community organizations, policy makers, and academics in the aforementioned sectors. We also hosted a group interview with newcomer mothers who had recently moved out of emergency shelter to verify and validate preliminary themes. The project was designed and implemented in partnership with the CAC and was successfully completed due to their commitment and collaborative approach. Analysis and the subsequent findings and implications were shared with the member group and developed jointly with the CAC.

## 5.2. Data Collection

Two groups of participants were interviewed for this study: newcomer mothers who were experiencing homelessness and service providers who worked with homeless and/or newcomer groups. Newcomer mothers were eligible to participate if they arrived in Canada within the last five years, self-identified as a mother, and were currently staying in an emergency or second-stage shelter. Service providers were eligible to participate if they had worked with homeless or newcomer populations. All participants were at least 18 years of age, spoke English, and provided signed consent.

Twelve newcomer mothers—the majority of whom were from visible minority groups—were recruited at two family shelters (of the five family shelters in Calgary). Mothers were invited to participate using posters displayed at the shelters, shared by agency staff, and word of mouth. Mothers who were interested in participating contacted the research team on their own or with the help of a service provider; a one-on-one interview time was set. All mothers received a $25 gift card to thank them for their time.

One group interview was held with 12 volunteers from the CAC and a second with four professionals working in the legal, immigration and homeless sectors. Both groups helped to clarify the major themes that were emerging as mothers shared their experiences.

Six mothers with previous, and in some cases very recent, experience with homelessness were included in a third group interview. The group interview was hosted by an immigration settlement agency and provided verification and expansion of the themes and findings from the individual interviews. An important part of community-engaged research is for the researchers to engage in iterative reflection throughout the research process [62]. This meant taking notes after every interview. Comments and discussions that stood out to the researchers, as well as initial themes, were recorded and used throughout the methodological and analytical process.

Advantages to the data collection included the representation of the diverse experiences of homelessness in newcomer women, from emergency shelters, second-stage shelters, and domestic violence shelters, as women may have stayed at multiple shelters during their time in Canada.

Furthermore, having a group interview with members of the target population for this study after the one-on-one interviews and analysis had been conducted helped to verify and frame themes accurately, adding to the validity of the study. Although the sample size of this study was small, the overlap in the experiences of the women strengthened the reliability of the themes reported. These themes were also confirmed by the service provider group interview.

*5.3. Thematic Analysis*

Interviews were audio-recorded and transcribed verbatim. Thematic coding was an iterative process that occurred throughout and after data collection [60]. The team began by grouping similar codes and highlighting recurring themes as they emerged [62]. One way of verifying internal validity in qualitative research is through triangulation of data, or collecting data using multiple sources [62]. The findings were triangulated in multiple ways: firstly, through a literature search on existing research on the experiences of homeless newcomer mothers (as presented above); secondly, using multiple sources of data including one-on-one and group interviews; thirdly, through individual and team iterative analysis of the transcripts; finally, through sharing findings with the CAC members for validation and verification. Each participant interview was anonymized by removing identifying information and assigning a participant code (P.1, P.2, etc.). Due to the difficulties involved in differentiating speakers in each focus group, we did not assign individual codes to those participants.

## 6. Results

All mothers had more than one child. Four entered Canada as refugee claimants but came in through the United States and so did not fall under the Safe Third Country Agreement due to irregular border crossing. Six entered Canada as immigrants, two women had expired visitor visas, and one had an expired temporary foreign worker visa. The remainder came directly to Canada as refugee claimants. The Safe Third Country agreement between Canada and the United States requires that refugees claim asylum in the first country where they arrive safely [67].

Three major themes emerged from the interviews: gendered and racialized pathways into homelessness, system failures that prevent exits from homelessness, and trauma experienced by the women both pre- and post-migration.

*6.1. Gendered and Racialized Pathways into Homelessness*

Two main subthemes emerged regarding newcomer women's pathways into homelessness: being a newcomer and being a mother.

*Being a Newcomer*

Participants in our study came from diverse backgrounds, with differing levels of education, language abilities and job experiences, differing experiences of trauma and different family backgrounds. All experienced difficulties accessing information about the immigration system that is mostly available through government websites or through conversation with immigration officers. All argued that it was typical to find complicated technical jargon rather than plain language information. This made it difficult when women were trying to access the rules before coming to Canada and when trying to move forward with their immigration applications after they arrived.

"I wish we had access to the right information before we got here. I know that information is online but they're not clear because if there is clearer information before anybody should embark on their journey it won't be stressful." (P5).

Women often were not fully aware of what was happening even as they were progressing through the system and making decisions that would affect their futures. For example, one woman had her passport taken away at the border by an immigration officer because she had arrived under the 'protected persons class'. However, the reason for her passport being confiscated was not provided until later—her passport had been her only form of identification and this left her confused and anxious. Another woman had a similar experience—when she found out she would have to give up her passport

at the border, she decided to enter Canada on a tourist visa. However, this created limitations in the supports that would have been available to her as a protected person. It also created long wait times as she tried to backtrack and re-apply as a refugee once she had already entered Canada.

Another expressed her confusion about the way the court hearing would work, saying "...I don't really understand...they just told us on such and such date of December that we'll have to do a hearing..." (P8). Our participants highlighted their misunderstanding of the processes and polices consistently and at multiple junctures. However, they accepted that they must follow the rules without actually knowing why they were doing it or how it would help their situation.

"I don't know how the system works here so my mind was open to like, ok whatever you guys have for us. We don't have a choice; we have to take it." (P9).

*Being a Mother*

For many participants, being a mother also limited their success in navigating the rules of the Canadian immigration system. Although this system is set up for a sponsored 'family' class, mothers (especially when acting as single parents) often faced issues because of their children. These included no access to childcare, added financial burden and custody battles. This experience was in stark contrast to their original intention of immigrating to Canada to get a better life for "all for their children" (P2).

Concerns about childcare, specifically for single mothers, were especially problematic for women who were 'claimants' (did not have permanent resident or refugee status in Canada), dealing with wait times or issues with paperwork. "For your kid to be even able to be taken care of, to be able to leave her at childcare or something you still have to have a status" (P3).

The lack of childcare also limited women's opportunities to go out and seek employment or to take language programs.

"There's no childcare or I've asked for childcare or where to find it where I wouldn't have to pay because there's no money right now so I can go do these things and I don't know where it is" (P1).

One mother said, "I won't be able to start work until my children resume school..." (P8). A major concern was the high cost of childcare and a lack of flexible hours. Getting to appointments on time to view houses, attend workshops, or meet with lawyers was also identified as being much more challenging with children: "Trying to be able to find a place while having children under foot. Those are the obstacles I think I found. Especially in the winter. Trying to go out to viewings with a child and the stroller in the snow or the cold." (P1).

Motherhood was also a unique barrier when addressing custody battles. These battles often necessitated navigating the Canadian legal system in addition to the immigration system. "We have to go through . . . court systems on top of homelessness and, and then looking for a place as well and, it's just hard. Because that gives more stress on top of a situation that's like this" (P1).

*6.2. System Failures*

Two major system failures were identified which affected mother's ability to exit homelessness related to jurisdictional issues and trying to change their 'status'. In the context of this study, we have defined system failures as elements of federal and provincial immigration policies that create barriers for newcomers to Canada due to their newcomer status and/or gender. System failures are failures in the design of the immigration system which is meant to guide and promote immigration and resettlement. However, in our study, system failures affect newcomer women's experiences in Canada and their housing status, leading them into homelessness. Furthermore, these are issues at the structural level, over which the women themselves have no control.

6.2.1. Jurisdictional Issues

Differences between municipal, provincial, and federal legislation created jurisdictional barriers that affected women's eligibility to access services. Women had to be in an emergency shelter to obtain basic needs such as food, housing, and health care. Inconsistent procedures, and 'discretionary

decision making' exacerbated their difficulties. Access to supports that fall within the jurisdiction of the provincial or municipal governments (health care, education, housing, social assistance) are dependent on decisions made at the federal level, often by federal immigration officers. One woman discussed her experience when her temporary foreign worker's permit expired after she and her children had a medical emergency. Upon leaving the hospital, she was faced with no employment and mounting bills from her and her children's hospital stay because she did not have a health care card. At every step, she had to access a different service or wait until someone could support her applications. She had to get special permission from several providers to reinstate her children's health card and to access legal supports.

"That's the time they took away the health card and I was getting bills, bills, bills... [eventually, with the help of a lawyer], the kids' health card was reinstated. [The lawyer] started my application [for permanent residency] I didn't have money for the application so I had to ask for money to pay for it . . . A nurse had to write a letter to Human Services [to get insurance to pay for medication for the kids]. But if I get sick, I will have to pay. I have a medical bill for over $1000, I just keep borrowing money to pay. I said I can't manage more, I'm due back to the doctor but I don't know what to do. I'm just hanging on here. It's really hard when you're here in Canada without family or without status." (P10).

Service providers also indicated that there was a lack of expertise at the frontline to deal with the complexities of these mother's situations. For example, an agency might specialize in supporting victims of domestic violence but have no legal expertise in immigration issues. Alternatively, they may have expertise in settlement supports but no expertise or resources to deal with family violence or the resultant trauma. Significant gaps occur in accessibility to wrap around or holistic supports available for women and children and the need to access multiple services for multiple issues leads to frustration, stress, and long wait times, where the only option for basic needs support is an emergency shelter.

### 6.2.2. Status

Women were often ineligible for government financial assistance or affordable housing, which typically falls within the mandate of provincial or municipal governments. This affected their ability to move forward in the settlement process. For refugee claimants, until their hearing to determine whether they can stay in Canada, they were unable to access supports from many settlement agencies who are funded to support immigrants but not refugees. This meant they could not take language classes, training or gain employment. This wait can be upwards of a year.

"The housing situation can be super challenging, depending on what their status is so that's a huge barrier to go through. Any government funded subsidy housing or case management program . . . I can't remember exactly the level of status that they require, but a lot of times they don't meet that so we can't house them" (Service Provider).

"A woman's status is not something we would turn her away for but we have to ask the question because it affects the other services we can provide or refer her to" (Service Provider).

Women's status in Canada was often dependent on their husband's sponsorship. Women discussed situations where their husbands or partners would keep important documents from them or limit their access to financial supports unless the woman would agree to stay or come back into the abusive home. One woman discussed her spousal sponsorship breakdown. Although legally, her husband, as her sponsor, was required to support her, he refused to do so. She was ineligible for provincial supports because of the existing sponsorship agreement and this led her into homelessness. Other women's husbands told them that they had applied for sponsorship, when they had not. When the relationships eventually broke down, this left the women without status, without sponsorship, and with limited options for support. Service providers discussed the limitations in such situations, that structural gaps keep these women trapped in homelessness and also discussed difficulties navigating multiple systems to try and support them.

"It is hard because we are funded to provide specific services and we live in constant fear of funding cuts, many of these women need more than we can give them." (Service provider)

## 7. Trauma

The theme of 'trauma' was pervasive in the interviews; the breadth and depth of trauma throughout the women's experiences highlighted the impact of both pre- and post-immigration trauma on the women's lives.

### 7.1. Pre-Migration Trauma

Pre-migration trauma was common and often a precipitating factor in a family's decision to relocate to Canada due to interpersonal violence or violence at the hands of persons of authority in their home country. Two participants described persecution against their husbands so that they could no longer stay in their home country.

"They had this issue in the office, it involved some of his employees engaging in fraud and because he was kind of a witness they started hunting him, like trying to kill him." (P9).

"We fled for safety because my husband had issues at work . . . they were after him so because of him it affected us so we moved from different states within our country but still we didn't feel safe, we had to move." (P5).

Other mothers cited gang violence, wars, police neglect and poorly resourced countries as reasons for bringing their children to Canada.

"They were killing people, so you have to run, until that war, 1990 war start, we lost everything in that war." (P7).

"Because back home you go to work, you work like five days per week and a payday a Friday, someone just robbed you, killed you, just like that. That's how they do it. They want to kill you, they just kick off your door and just kill you." (P10).

Several of the mothers spoke of fears for the safety of their children, "I have to leave situation because my life was at stake, my children's life was at stake." (P8).

A common reason that women left their countries and decided to migrate to Canada was due to familial or intimate partner violence.

"I won't go to Jamaica. Reason why? Those kids, their dad, he almost let me lose my life back home before I came here, we were fighting, fighting, fighting. I went to work, he come at my workplace, fighting, I had to call the cops on him. So what I have to do [is] to avoid him, I have to walk with knife." (P10).

### 7.2. Post-Migration Trauma

Post-migration trauma for mothers in our study occurred at both the individual and institutional levels. Post-migration trauma is related to system failures as it is often outside of the control of women, however, it is distinct in our study in that it occurs as a result of the system failures. Because women felt powerless to address barriers in the Canadian immigration system, they experienced trauma due to the impact of these system failures on their ability to traverse the immigration process as newcomers to Canada.

Four mothers first immigrated to the United States but quickly left due to fear of newly implemented immigration policies in the US, long wait times or refusal of services, forcing them to cross illegally over the Canadian border into Quebec. They were immediately arrested and detained. One mother recounted the extremely stressful experience of crossing the border in the winter:

"We were detained in an office we were searched. They asked us some questions like where we coming from, how much do we have, why are we here and they took us to the immigration office, so there was like a long night, long and cold cause it was very cold when we came in... We were kept there for hours, it was so stressful for me cause I was just pregnant, it was my early pregnancy so I was so stressed out." (P9).

The Canadian immigration office was often cited as being incredibly difficult to work with. "The support I need, nobody can help me. That is my main problem here, I just want my paperwork

and just get out [of shelter]" (P10). Additional services that mothers expressed wanting or needing were legal counseling and legal representation, services for their children, income supports and housing. Mothers cited depleted energy and mental health struggles due to the constant stress. "[The immigration office] require evidence to prove the abuse—I needed a protection order… But had no direction or guidance on what is needed to get that". (P8).

One mother had her child taken away from her by her spouse when she had to travel for immigration papers and was left with no way to get her child back. "He cut off all communication. So I was worried, I mean he has my baby" (P7). One mother experienced complete abandonment "I came here [shelter] because my husband abandoned me. He take my children and never allow my children to talk with me. And I never see them." (P2).

When mothers felt like they had been abandoned, they no longer had the energy to look for support.

"Maybe you are so in panic, you are shocked there is nothing you can do, you don't even want to read anything, frustration and it can be pretty difficult." (P3).

"That was very, very confusing, very challenging and it was like we were in the wrong place. That's how I felt because at that time we were like oh should we go back [home]. It was so stressful." (P5).

## 8. Discussion

Our study used a critical lens [68,69], allowing us to focus on the structural barriers using a gendered and cultural framework. Focusing specifically on culture or gender would result in a singular view of the newcomer mother experience of homelessness. Other qualitative work in this area highlights the need to examine the intersectional factors of both migration and gender, which combine to affect the vulnerability of newcomer women to housing instability and homelessness [70]. Our approach recognizes the "disadvantage of immigrant women in relation to the dynamics of violence: ethnicity, immigrant status, and interacting gender inequalities" [71], (p. 2). Our study also provides a unique examination of the structural factors affecting newcomer women's homelessness in Canada, helping to fill a substantial gap in the literature.

Although internationally recognized for welcoming attitudes and processes for immigrants and refugees, our results reveal that the Canadian immigration system has serious flaws that negatively affect the very people the system is meant to protect. The system failures experienced by the women in our study are the result of immigration policy that is not responsive to the needs of mothers who are or have experienced violence either before or after arriving in Canada. Feminist and anti-racist scholars have argued that while the Act makes a commitment to anti-racist and sexist discrimination principles such as offering a "safe haven to persons with a well-founded fear of persecution based on race, religion, nationality, political opinion", trying to reunite families, and offering a safe place for those fleeing conflict and violence, it does not do so in practice. In other words, the Act allows discrimination and sexism to continue [72].

Similar to Stephens and colleagues [73], who noted the complex relationship between factors such as immigration status, access to resources, policy, and personal characteristics (e.g., presence of children or a partner) that affect the vulnerability of women to housing instability and homelessness, our study found that both interpersonal and structural crises compound to leave some mothers and their children trapped in a cycle of homelessness and violence that is difficult to escape. Immigration policy and processes have been developed with little to no recognition of gender and cultural differences, the trauma that many newcomer mothers face, and sponsorship breakdowns. They have been developed within neoliberal priorities of economic success and independence, and women with precarious status are penalized because they are not able to meet the standards of 'good' citizenship, self-sufficiency and paid work. The system failures resulted in their further marginalization.

Interventions to support or address the issues mothers face operate in a siloed fashion—systems are slow to communicate with each other, requiring proof of breakdown from each other, all the while suspending the resettlement process. Furthermore, a gap exists between the federal government and the provincial and municipal services. The Government of Canada mandates settlement services.

However, this support is usually short term and does not continue to support beyond a one-year timeframe [74]. This leaves many provincial and municipal governments and community-based service providers with the burden of supporting immigrants and refugees, while not adequately funded or mandated to do so. This leaves many newcomer mothers and their children without support, without status, and without a home. Similar results have been found internationally, with a small study on gender-based violence and migrant women's homelessness in Ireland reporting that homelessness in newcomer women was strongly connected to structural factors such as socioeconomic status, which was, in turn, linked to immigration status and financial independence [75]. However, our study takes this analysis further to examine the structural factors affecting newcomer women's homelessness themselves, rather than only examining the impact that they have on this population.

The Act and subsequent processes were created to promote success for newcomers to Canada with the underlying neoliberal value of economic growth for Canada, rather than humanitarian values. These values to do not align with the complexities of race or motherhood. Instead, immigration policy excludes and discriminates against newcomer mothers and causes further trauma to these women and their children.

There are two major policy and practice implications of this study. Firstly, funding models for supporting newcomers to Canada should be adaptable; additional federal and provincial funding should be provided to settlement agencies to support all newcomers to Canada for longer than one year, including refugee claimants. If adequately funded, these agencies could provide a centralized 'hub' with collaborative and comprehensive expertise for addressing the complexities of the mothers' experience. Secondly, wrap-around and holistic supports for violence, trauma, housing and settlement should include access to free or affordable childcare, legal supports and mechanisms for changes in sponsorship when this relationship breaks down. Settlement supports should be grounded in trauma-informed practice, recognizing and responding to the multitude and diverse experiences these women face pre and post-migration. A 'rights-based' approach that recognizes and responds to gender and cultural diversity and intersectionality challenges 'one-size-fits-all' approaches and calls for a re-examination and re-framing of economic-focused federal immigration policy that results in structural violence and structural discrimination.

As there is still limited research and understanding on the intersectional experiences of newcomer women's homelessness, further research in this area is necessary at both the academic and practice levels. Future studies should examine the complexities of immigrant status and immigration class on newcomer women's homelessness, how the presence of children or a partner can affect newcomer women's homelessness, and how policy shifts impact these experiences (for example, the Canadian government's acceptance of a high number of Syrian refugees and the adapted policy measures for providing resettlement services).

## 9. Strengths and Limitations

Our study had several important strengths and limitations. Although this was a small qualitative study, the intersectional analysis allowed us to move beyond individual-level or interpersonal issues to the examination of Canadian structures, values, and beliefs that create systems which perpetuate trauma for newcomer mothers. In addition, our sample included the diverse experiences of newcomer mothers, with women from various ethnic, economic, and cultural backgrounds, and with unique immigration stories. Our study also benefitted from the community-based approach, which allowed for guidance on the development of the research questions and interview tools, support for recruitment, and verification of the emerging themes. Our study may have been limited by our recruitment approach; it is possible that other newcomer mothers experience homelessness but were not staying in shelters. Further, participation was limited to those who spoke conversational English; we may have unintentionally limited the scope of our study and missed additional barriers faced by these women.

## 10. Conclusions

Our study revealed broad and deep complexities in mothers' experiences within Canada's immigration system. Within their stories, systemic failures have been exposed. These failures create and exacerbate vulnerabilities for these women and their children. Newcomer women and children with uncertain or precarious status are vulnerable to the sustained experiences of homelessness and its deleterious effects, because they are at high risk for trauma with limited access to resources when they are in crisis. The issue originates in federal policy, indicating that structural violence and discrimination are embedded in Canadian immigration law, perpetuating mothers' reliance on and subsequent exclusion from public systems.

Pathways into homelessness are gendered and racialized, and significant policy changes are necessary to address these failures. Canada must critically review its immigration policy to focus on supporting vulnerable newcomers to Canada, rather than maintaining the existing economic policies that result in racist and patriarchal approaches. Removing structural barriers to success for newcomer mothers to Canada can promote healthy and expedited resettlement processes, improve the health and well-being of vulnerable families and make Canada the inclusive and welcoming country it purports to be. However, a critical examination of current legislation and its implications on service delivery should not be done in the absence of an equitable partnership with service providers and the mothers themselves.

**Author Contributions:** Conceptualization, K.M. and R.T. methodology, K.M. and R.T. writing—original draft preparation, K.M.; R.T.; S.B. and K.R.; writing—review and editing, K.M.; R.T.; S.B. and K.R. All authors have read and agreed to the published version of the manuscript.

**Funding:** This work was supported by the Social Sciences and Humanities Research Council of Canada under grant 430-2017-00187. This is to acknowledge that no financial interest or benefit to the authors has arisen from the direct applications of our research. Katrina Milaney is an Associate Professor in Community Rehabilitation and Disability Studies at the University of Calgary. Katrina is a qualitative and mixed-methods researcher who uses critical theory frameworks to study social vulnerabilities related to disability, homelessness, gender, culture, domestic violence, and mental health. Her primary interests revolve around political and economic ideology and their impact on public systems and service delivery. Katrina is a Distinguished Policy Fellow and is the recipient of a University of Calgary Peak Scholar award, O'Brien Institute Societal Impact Award and the Cumming School of Medicine Distinguished Achievement for Social Accountability. She was recently named one of the top 20 Most Compelling Calgarians for 2020.

**Access to Data:** This was a qualitative study. The Ethics Review Board decisions require that only the authors have access to the transcripts.

**Conflicts of Interest:** The authors declare no conflict of interest.

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
