# Peer review of "Welcome to Canada: Why Are Family Emergency Shelters ‘Home’ for Recent Newcomers?"

_societies, doi:10.3390/soc10020037_

Round 1

Reviewer 1 Report

The article focuses on describing the experiences of vulnerable immigrant women in Canada within the context of neoliberal government structures and policies that exacerbate the challenges that the immigrant women face.  The article’s stated goals are “ firstly, to examine newcomer mothers’ experiences of 185 homelessness to identify structural imbalances and gaps; secondly, to understand how structural barriers 186 influence violence and trauma; and finally, to recommend changes to  both policy and service delivery to 187 address these structural issues.”  Through in-depth interviews with 12 immigrant mothers, 12 volunteers from the Community Advisory Committee, and four professionals from legal, immigration, and homeless organizations, the author finds three major themes:  “gendered and racialized pathways into 283 homelessness, systems failures that prevent exits from homelessness, and individual and institutional 284 trauma experienced by the women both pre and post-migration.”

Overall, the article is an informative and interesting examination into the experiences of this class of immigrants (lower income, culturally distinct, mothers) and the multi-level challenges and barriers that they face in Canada’s immigration system and society as a whole.  The author does a very good job in first, summarizing the general structure and policy frameworks of Canada’s immigration system, then presenting the ethnographic interview data of the sample group of immigrant women to highlight how disadvantages, challenges, and barriers at the individual, community, and institutional levels that the women face as the try to build a safe and comfortable life in Canada.  The author also does an excellent job of presenting possible policy changes and other potential fixes to the problem (lines 498-509).

First, I would like to point out some structural/grammatical errors:

  • In line 28, the author mentions the “economic class” type of immigrant, but does not define or describe that until line 75; it needs to be defined earlier in the paper
  • Lines 37-40 is a run-on sentence
  • The two paragraphs from lines 135-157 are almost identical in content and therefore, redundant

Second, following are my more substantive comments.  One critique is that throughout the article, the author seems to conflate immigrant status with racial identity.  Specifically, several times in the article, the author mentions race as an important factor in Canada’s immigration process (lines 13, 46, 47, 55, 57, 58, etc.).  However, as the article proceeds, there is no actual discussion or analysis of race or ethnicity, either in terms of specific government policies that treat groups or individuals differently according to their racial or ethnic identity, or even in presenting the ethnographic data of the interview respondents.  Instead, the author seems to subsume race/ethnicity in the immigrant status of the respondents or the larger population of immigrant women in Canada.  In other words, despite noting the salience of race/ethnicity in the introduction, the author conflates these identities with immigrant status and assumes that being an immigrant, a “newcomer” also means being a racial/ethnic minority member, but without any discussion or analysis of how the situation of the respondents or the institutional structure of Canada’s immigration policies have been or continue to be racialized in practice.  This is a very important shortcoming that requires substantial revision.

Second, in presenting the data about the challenges that the immigrant women respondents face, the author separates “system failures that prevent exits from homelessness” and “. . . institutional trauma experienced by the women . . . post-migration” as two separate analytical themes in the study’s findings (lines 282-284).  However, in the subsequent data results and presentation, there again seems to be a conflation of the two and ultimately, there does not seem to be any substantive difference between the two as they both refer to organizational policies and bureaucratic inefficiencies that compound the difficulties that the immigrant women face in improving their lives through trying to access and navigate through the governmental framework.  In other words, the article fails to show how “system failures that prevent exits from homelessness” and “. . . institutional trauma experienced by the women . . . post-migration” are different from each other.

Ultimately, the article shows promise and focuses on important issues but requires substantial reconceptualization and reorganization.

Author Response

Reviewer #1 Comments

  1. First, I would like to point out some structural/grammatical errors:
    1. In line 28, the author mentions the “economic class” type of immigrant, but does not define or describe that until line 75; it needs to be defined earlier in the paper.

Thank you for this comment. We have moved the paragraph defining the classes of immigration in Canada earlier in the Introduction. 

  1. Lines 37-40 is a run-on sentence

We have split this sentence into two sentences.

  1. The two paragraphs from lines 135-157 are almost identical in content and therefore, redundant

Thank you for this comment. We have restructured these paragraphs to avoid redundancy of information. 

  1. Second, following are my more substantive comments.  One critique is that throughout the article, the author seems to conflate immigrant status with racial identity.  Specifically, several times in the article, the author mentions race as an important factor in Canada’s immigration process (lines 13, 46, 47, 55, 57, 58, etc.).  However, as the article proceeds, there is no actual discussion or analysis of race or ethnicity, either in terms of specific government policies that treat groups or individuals differently according to their racial or ethnic identity, or even in presenting the ethnographic data of the interview respondents.  Instead, the author seems to subsume race/ethnicity in the immigrant status of the respondents or the larger population of immigrant women in Canada.  In other words, despite noting the salience of race/ethnicity in the introduction, the author conflates these identities with immigrant status and assumes that being an immigrant, a “newcomer” also means being a racial/ethnic minority member, but without any discussion or analysis of how the situation of the respondents or the institutional structure of Canada’s immigration policies have been or continue to be racialized in practice.  This is a very important shortcoming that requires substantial revision.

Thank you for this comment. We have clarified the relationship between immigration status and racial identity. Indeed, these are two distinct concepts. However, they are also linked. The Canadian Council for Refugees (2000) notes that “in Canada, where open expression of racist ideas is generally not tolerated, hostility towards newcomers serves as an outlet for the expression of underlying racist sentiments” (para 2). Furthermore, limiting discussion of race and racism to situations where there are differences in genetically-determined, phenotypical characteristics between groups excludes situations where race is perceived to be an issue and where racism may still occur (Richmond, 2001). The International Convention on the Elimination of All Forms of Racial Discrimination (1965) defines racial discrimination as “any distinction, exclusion, restriction or based on race, colour, descent, or national or ethnic origin” (Article 1). All newcomers to Canada are thus potentially subject to racism, regardless of their status as a racial or ethnic minority. 

  1. Second, in presenting the data about the challenges that the immigrant women respondents face, the author separates “system failures that prevent exits from homelessness” and “. . . institutional trauma experienced by the women . . . post-migration” as two separate analytical themes in the study’s findings (lines 282-284).  However, in the subsequent data results and presentation, there again seems to be a conflation of the two and ultimately, there does not seem to be any substantive difference between the two as they both refer to organizational policies and bureaucratic inefficiencies that compound the difficulties that the immigrant women face in improving their lives through trying to access and navigate through the governmental framework.  In other words, the article fails to show how “system failures that prevent exits from homelessness” and “. . . institutional trauma experienced by the women . . . post-migration” are different from each other.

Thank you for this comment. We believe that systems failures and trauma are two different themes that emerged from the data. Firstly, systems failures refers to macro-level factors built into federal and provincial immigration policies (jurisdictional issues and immigration status) that affect women’s experiences and over which the women have no control. These were seen by the researchers to be failures in the design of the system, which in theory was designed to promote immigration and resettlement, but in practice has many barriers to achieving this goal.

Secondly, post-migration trauma (and thus the institutional trauma) were seen to be the result of the systems failures. This section showed the impact of these the systems failures on women’s ability to traverse the immigration process as newcomers to Canada. We have clarified this distinction in the text.

Reviewer 2 Report

“Welcome to Canada: Why are family emergency shelters ‘home’ for recent newcomers”

General Comments:

This is an important topic and a useful paper focusing on “newcomer mothers’ experiences of homelessness to identify structure imbalances and gaps…[and]… understand how structural barriers influence violence and trauma.” (p. 5). This is an important research topic and fills, in part, a major gap in the Canadian literature. Also this study has policy implications. My overall assessment is that this paper would make an important contribution to the literature, as well as an informative reading for the audience of the Societies, but only if some minor revisions are undertaken by the author(s).    

Specific Comments:

  1. Introduction (pp. 2-3): Expand on why this research topic is important. Provide a stronger rationale for choosing recent newcomer mothers with current or recent experiences with homelessness in the City of Calgary (Canada). What makes this group of newcomers so unique in the City of Calgary (“new port of entry” for immigrants and refugees in Canada)? Any research question(s) guiding this study? If yes, please add them to the introduction of this paper. In sum, there is room for a stronger rationale/justification of the research topic, the study group, and the study area.
  2. “Background” (pp. 3-5): The author(s) are familiar with the literature. However, I recommend the author(s) to expand the review of the literature on immigrant and refugees settlement and integration experiences in Canada’s major metropolitan areas, including the City of Calgary a new destination for newcomers (e.g., Who are the most recent immigrants and refugees to the country and where do they settle in the country by province and major cities of settlement (ports of entry”); major barriers and challenges they face in accessing settlement services, housing and jobs; expand on discrimination encountered by newcomers in Canada and on the main “forces” at play (role and impact of “urban gatekeepers”……)
  3. “Methodology” (p. 5-6): The author(s) made good use of the sources used (e.g., qualitative study/ethnographic research; Critical Social Theory (CST); Community-engaged research design (CER)…). However, I suggest the author(s) expands on the major advantages and limitations of the main source used to collect data as well as on the small sample/sampling strategies. Also provide more information on how volunteers from the CAC were initially identified and selected. Also provide more information on the main criteria to select two shelters in Calgary (how many shelters do we have in Calgary?).
  4. “Results” (pp. 7-11). I enjoyed reading this section of the paper. However, the balance of description and interpretation must be shifted toward the latter. How do the results reported in this study corroborate (or not) other studies in Canada?
  5. “Discussion” and “Conclusion” (pp. 11-13). Expand on “areas for further research” as well as on the major policy implications of this study.

Author Response

Reviewer #2 Comments

General Comments: This is an important topic and a useful paper focusing on “newcomer mothers’ experiences of homelessness to identify structure imbalances and gaps…[and]… understand how structural barriers influence violence and trauma.” (p. 5). This is an important research topic and fills, in part, a major gap in the Canadian literature. Also this study has policy implications. My overall assessment is that this paper would make an important contribution to the literature, as well as an informative reading for the audience of the Societies, but only if some minor revisions are undertaken by the author(s).    

  1. Introduction (pp. 2-3): Expand on why this research topic is important. Provide a stronger rationale for choosing recent newcomer mothers with current or recent experiences with homelessness in the City of Calgary (Canada). What makes this group of newcomers so unique in the City of Calgary (“new port of entry” for immigrants and refugees in Canada)? Any research question(s) guiding this study? If yes, please add them to the introduction of this paper. In sum, there is room for a stronger rationale/justification of the research topic, the study group, and the study area.

Thank you for this comment. We have strengthened the rationale in the introduction section to include the lack of research in this area, the importance of a gendered analysis, and the research question guiding this qualitative study.

  1. “Background” (pp. 3-5): The author(s) are familiar with the literature. However, I recommend the author(s) to expand the review of the literature on immigrant and refugees settlement and integration experiences in Canada’s major metropolitan areas, including the City of Calgary a new destination for newcomers (e.g., Who are the most recent immigrants and refugees to the country and where do they settle in the country by province and major cities of settlement (ports of entry”); major barriers and challenges they face in accessing settlement services, housing and jobs; expand on discrimination encountered by newcomers in Canada and on the main “forces” at play (role and impact of “urban gatekeepers”……)

Thank you for this important content. We have added a section to the background of the paper describing immigration trends in Canada and some of the barriers experienced by newcomers to Canada.

  1. “Methodology” (p. 5-6): The author(s) made good use of the sources used (e.g., qualitative study/ethnographic research; Critical Social Theory (CST); Community-engaged research design (CER)…). However, I suggest the author(s) expands on the major advantages and limitations of the main source used to collect data as well as on the small sample/sampling strategies. Also provide more information on how volunteers from the CAC were initially identified and selected. Also provide more information on the main criteria to select two shelters in Calgary (how many shelters do we have in Calgary?).

Thank you for this comment. We have clarified the recruitment of the CAC and the selection of the two shelters in Calgary. We have also added some advantages/limitations of the methods used as well as the sample size for the study.

  1. “Results” (pp. 7-11). I enjoyed reading this section of the paper. However, the balance of description and interpretation must be shifted toward the latter. How do the results reported in this study corroborate (or not) other studies in Canada?

Thank you for your comment. We feel that the Results section of the study should highlight only the results of the current study and instead, the Discussion section should be used to examine the results of our study in the context of the broader literature. We have added several such points to the Discussion section of the paper.

  1. “Discussion” and “Conclusion” (pp. 11-13). Expand on “areas for further research” as well as on the major policy implications of this study.

Thank you for this comment. We have highlighted several policy implications in the discussion section to make our aforementioned recommendations clearer. We have also added a short paragraph on areas for further research.

Round 2

Reviewer 1 Report

The edits and clarifications are greatly appreciated and improve the quality of the paper.  Regarding my previous critique about the relatively lack of differentiation between “system failures that prevent exits from homelessness” and “. . . institutional trauma experienced by the women . . . post-migration,” while they are clearly related, I still think that the paper would be improved by a more detailed explanation of how they are qualitatively distinct from each other.  Nonetheless, I am mostly satisfied with how their discussion was changed and clarified.

Unfortunately, my first critique about the conflation of immigrant status and racial/ethnic identity still stands.  Yes they are inevitably linked and I agree that the two are often used together as mechanisms of discrimination and exclusion, but I maintain that they are often qualitatively different ways in which newcomers are potentially subject to racism.  For example, an immigrant from an European country who is racially White is likely to be treated and welcomed differently (and most likely more favorably) than an immigrant from an African, Asian, or Latin American country who is racially non-White. 

As such, among the immigrant population, there are structural advantages and disadvantages associated with racial/ethnic identity (or at least perceived racial/ethnic identity) that lead to immigrants from racially White backgrounds generally being more welcomed than immigrants from non-White backgrounds.  This ongoing predominance of race and ethnicity interacts with immigrant status to affect the outcomes of newcomers to nations such as Canada, but I maintain that the paper needs a more detailed and convincing analysis of how race/ethnicity might operate distinctly from immigrant status. 

If it turns out that the vast majority of respondents in this study are in fact non-White, than the paper should just state that upfront, acknowledge that its sample population is limited to non-White immigrants, and remove or at the very least deemphasize its goal of analyzing race/ethnicity as an important factor in Canada’s immigration process.

Author Response

Thanks very much to the reviewers for their thoughtful comments 

Reviewer comment: The edits and clarifications are greatly appreciated and improve the quality of the paper.  Regarding my previous critique about the relatively lack of differentiation between “system failures that prevent exits from homelessness” and “. . . institutional trauma experienced by the women . . . post-migration,” while they are clearly related, I still think that the paper would be improved by a more detailed explanation of how they are qualitatively distinct from each other.  Nonetheless, I am mostly satisfied with how their discussion was changed and clarified.

Response: We have edited lines 487-492 to say: “Post-migration trauma for mothers in our study, occurred at both the individual- and institutional-levels. Post migration trauma is related to system failures as it is often outside of the control of women, however it is distinct in our study in that it occurs as a result of the system failures (as described above). Because women felt they were powerless to address the barriers in the Canadian immigration system, they experienced trauma due to the impact of these systems failures on their ability to traverse the immigration process as newcomers to Canada.”

Reviewer comment: Unfortunately, my first critique about the conflation of immigrant status and racial/ethnic identity still stands.  Yes they are inevitably linked and I agree that the two are often used together as mechanisms of discrimination and exclusion, but I maintain that they are often qualitatively different ways in which newcomers are potentially subject to racism.  For example, an immigrant from an European country who is racially White is likely to be treated and welcomed differently (and most likely more favorably) than an immigrant from an African, Asian, or Latin American country who is racially non-White.

As such, among the immigrant population, there are structural advantages and disadvantages associated with racial/ethnic identity (or at least perceived racial/ethnic identity) that lead to immigrants from racially White backgrounds generally being more welcomed than immigrants from non-White backgrounds.  This ongoing predominance of race and ethnicity interacts with immigrant status to affect the outcomes of newcomers to nations such as Canada, but I maintain that the paper needs a more detailed and convincing analysis of how race/ethnicity might operate distinctly from immigrant status.

If it turns out that the vast majority of respondents in this study are in fact non-White, than the paper should just state that upfront, acknowledge that its sample population is limited to non-White immigrants, and remove or at the very least deemphasize its goal of analyzing race/ethnicity as an important factor in Canada’s immigration process.

Response: Thank you for asking for this clarification it is an important one - the majority of our participants were indeed non-white, we have added a sentence about this to the methods.  However – I am not clear how to address this comment in particular “remove or at the very least deemphasize its goal of analyzing race/ethnicity as an important factor in Canada’s immigration process.” We believe that because women who are non-white immigrants would face exacerbated racism and discrimination – that race/ethnicity is even more important to discuss in the immigration process than if our participants were white immigrants.

We have tried to address this by choosing to use the language structural discrimination instead of racism and have added this…

Lines 70-75 have been edited to say “Racism and discrimination are terms that are often used interchangeably, in this study we choose to use the term discrimination as it has been defined as “unequal treatment based on group membership.” Like structural violence, structural discrimination is often embedded in policies, institutions and subsequent practices. Although not every newcomer to Canada is a member of a racial or ethnic minority, they still experience the impacts of racism and racial discrimination…”